# Cholesterol Metabolic Profiling of HDL in Women with Late-Onset Preeclampsia

**DOI:** 10.3390/ijms241411357

**Published:** 2023-07-12

**Authors:** Tamara Antonić, Daniela Ardalić, Sandra Vladimirov, Aleksandra Zeljković, Jelena Vekić, Marija Mitrović, Jasmina Ivanišević, Tamara Gojković, Jelena Munjas, Vesna Spasojević-Kalimanovska, Željko Miković, Aleksandra Stefanović

**Affiliations:** 1Department of Medical Biochemistry, Faculty of Pharmacy, University of Belgrade, Vojvode Stepe 450, 11000 Belgrade, Serbia; tamara.antonic@pharmacy.bg.ac.rs (T.A.);; 2The Obstetrics and Gynecology Clinic Narodni Front “Narodni Front”, Kraljice Natalije 62, 11000 Belgrade, Serbia

**Keywords:** ApoA-I, biochemical indices, CETP, LCAT, lipids, NCS_HDL_, phytosterols, PON1, preeclampsia, sterols

## Abstract

A specific feature of dyslipidemia in pregnancy is increased high-density lipoprotein (HDL) cholesterol concentration, which is probably associated with maternal endothelium protection. However, preeclampsia is most often associated with low HDL cholesterol, and the mechanisms behind this change are scarcely explored. We aimed to investigate changes in HDL metabolism in risky pregnancies and those complicated by late-onset preeclampsia. We analyze cholesterol synthesis (cholesterol precursors: desmosterol, 7-dehydrocholesterol, and lathosterol) and absorption markers (phytosterols: campesterol and β-sitosterol) within HDL particles (NCS_HDL_), the activities of principal modulators of HDL cholesterol’s content, and major HDL functional proteins levels in mid and late pregnancy. On the basis of the pregnancy outcome, participants were classified into the risk group (RG) (70 women) and the preeclampsia group (PG) (20 women). HDL cholesterol was lower in PG in the second trimester compared to RG (*p* < 0.05) and followed by lower levels of cholesterol absorption markers (*p* < 0.001 for campesterol_HDL_ and *p* < 0.05 for β-sitosterol_HDL_). Lowering of HDL cholesterol between trimesters in RG (*p* < 0.05) was accompanied by a decrease in HDL phytosterol content (*p* < 0.001), apolipoprotein A-I (apoA-I) concentration (*p* < 0.05), and paraoxonase 1 (PON1) (*p* < 0.001), lecithin–cholesterol acyltransferase (LCAT) (*p* < 0.05), and cholesterol ester transfer protein (CETP) activities (*p* < 0.05). These longitudinal changes were absent in PG. Development of late-onset preeclampsia is preceded by the appearance of lower HDL cholesterol and NCS_HDL_ in the second trimester. We propose that reduced capacity for intestinal HDL synthesis, decreased LCAT activity, and impaired capacity for HDL-mediated cholesterol efflux could be the contributing mechanisms resulting in lower HDL cholesterol.

## 1. Introduction

Preeclampsia is one of the most severe pregnancy complications, with an estimated incidence of 4.6%, although wide variations are recorded across regions [1]. It is defined as new-onset hypertension diagnosed after the 20th week of gestation with subsequent proteinuria or some other signs of end-organ dysfunction, such as thrombocytopenia, damage to liver function, development of renal failure, pulmonary edema, or new-onset cerebral or visual disturbances [2]. The latest approach regarding preeclampsia implies that this pregnancy complication is manifested by two different entities—early-onset preeclampsia, which develops before the 34th week of gestation, and late-onset preeclampsia, which tends to develop after the 34th gestational week [3].

Both early- and late-onset preeclampsia are thought to develop in two stages. The first stage includes poor perfusion of the placenta and consequent placental dysfunction. In response to the inadequate blood supply to the placenta and consequent hypoxia, oxidative stress develops, and the production of free radicals and inflammatory cytokines by the placenta increases. These molecular mediators of oxidative stress and inflammation lead to generalized maternal endothelial dysfunction in the second stage of the disease [4,5]. However, the key difference between early- and late-onset preeclampsia lies in the cause and time of placental malperfusion and dysfunction [4]. In the early-onset syndrome, the first stage results from poor placentation due to the shallow remodeling of spiral arteries during the first half of pregnancy. Incomplete remodeling of spiral arteries underlies abnormal uteroplacental perfusion, leading to placental oxidative stress and causing the hypersecretion of inflammatory and antiangiogenic factors into the maternal circulation [4]. In women with late-onset preeclampsia, the maturating placenta outgrows the uterus capacity and the supporting maternal functions. In this case, the placenta also becomes under-perfused, with restricted intervillous perfusion, causing placental stress at a later gestational age [4,5]. Placental overgrowth seems to be particularly associated with maternal obesity and large placentas [4]. Hence, both pathways cause placental hypoperfusion and dysfunction in the first stage, leading to the clinically recognized maternal syndrome of preeclampsia in the second stage of disease development. Early-onset syndrome is associated with adverse maternal and neonatal outcomes, such as reduction in placental volume, intrauterine growth restriction (IUGR), abnormal uterine and umbilical artery Doppler evaluation, low birth weight, multiorgan dysfunction, and perinatal death [5,6]. On the other hand, the late-onset disorder is usually associated with a larger placental volume, normal uterine and umbilical artery Doppler evaluation, a normally grown baby with no signs of growth restriction, and significantly more favorable outcomes for the mother and fetus [5,6]. However, late-onset preeclampsia has a higher incidence (2.7% vs. 0.38%) [5,6].

Cardiovascular disease and preeclampsia seem to share several risk factors, such as endothelial dysfunction, hypertension, diabetes, and low-grade systemic inflammatory response [3]. Maternal risk factors, such as chronic vascular diseases, obesity, autoimmune diseases, insulin resistance, and chronic kidney or liver diseases, seem to affect both stages of the disease development. These factors not only slow down the process of placentation (stage 1) but also increase the maternal vascular sensitivity to factors released by the placenta (stage 2). The increased sensitivity of the maternal endothelium to the factors released by the placenta is based on a chronic low-grade inflammation present in these women even before pregnancy [4]. Disturbances in lipid and lipoprotein metabolism are already acknowledged as key contributors to atherosclerosis and further cardiovascular disease development. Hence, modulators of lipid metabolism emerge as compelling factors that could be important in preeclampsia development [7].

Higher plasma lipid levels alongside significant lipid metabolism alterations are typical features of normal pregnancy [8]. Higher concentrations of total, low-density lipoprotein (LDL) cholesterol, and triglycerides are recorded in pregnant compared to nonpregnant women [8]. A specific feature of dyslipidemia in pregnancy is increased high-density lipoprotein (HDL) cholesterol concentration [8]. Some investigators hypothesized that an increase in HDL cholesterol concentrations during the second trimester of pregnancy is associated with maternal endothelium protection, and results of previous studies indicated that this change in HDL cholesterol concentration usually does not occur in women with preeclampsia [9].

Although variations in HDL cholesterol concentration during both a healthy pregnancy and a pregnancy with complications are well known, the mechanisms behind these changes have been scarcely explored. The maturation of HDL is a complex process comprising cholesterol uptake, esterification, and exchange with other lipoproteins. Thus, variations in the cholesterol content of HDL are observed as indicators of its metabolic transformation. Since non-cholesterol sterols (NCS) follow the same metabolic pathways as cholesterol itself, they are widely accepted as valid plasma markers of overall cholesterol metabolism, encompassing its endogenous synthesis (cholesterol precursors) and intestinal absorption (phytosterols) [10]. Recently, it was proposed that non-cholesterol sterols measured in isolated plasma HDL fraction (NCS_HDL_) could also represent valuable indicators of HDL metabolism [11]. Specifically, NCS_HDL_ reflects the proportion of both endogenously synthesized and exogenous absorbed cholesterol, which are carried by HDL particles. Accordingly, the determination of NCS_HDL_ could be useful for a better understanding of HDL particle metabolism during pregnancy with or without complications.

A comprehensive analysis of changes in HDL particles during pregnancy also requires the determination of the activity of two enzymes: lecithin–cholesterol acyltransferase (LCAT) and cholesterol ester (CE) transfer protein (CETP). By esterifying free cholesterol (FC) on the surface of HDL, LCAT enables the partition of the newly formed CEs into the core of the particle and the formation of mature spherical HDL particles [12]. CETP catalyzes the reaction in which CEs are exchanged for triglycerides between triglyceride-rich very-low-density lipoproteins (VLDL) and HDL and LDL [13]. Recent research has also provided evidence that modulation of CETP activity can affect the balance between cholesterol synthesis and absorption markers [14]. Pregnancy is usually characterized by a significant increase in the concentration of triglycerides, as well as by a rise in CETP activity [8]. Although changes in CETP activity have been described in detail in physiological pregnancy [8] and in some maternal and fetal complications [15], the literature lacks data on changes in the activity of this HDL key remodeling enzyme in women with preeclampsia.

Lastly, having in mind the complexity of HDL structure–function interactions [16], it should be investigated whether changes in NCS_HDL_ during pregnancy go along with alterations of key HDL functional proteins. Apolipoprotein A-I (ApoA-I) is required for normal HDL maturation and metabolism [16]. By interacting with different receptors and transporters, apoA-I stimulates the reverse transport of cholesterol. ApoA-I has been recognized as an LCAT cofactor, additionally promoting cholesterol efflux from peripheral tissues [17]. An HDL-associated esterase, responsible for the gross antioxidant properties of HDL, paraoxonase 1 (PON1), binds to HDL via interaction with ApoA-I and promotes enzyme stability [18]. On the other hand, some researchers have recently concluded that PON1 might contribute to cholesterol homeostasis by improving the capacity of HDL to mediate cholesterol efflux [19,20].

This study aimed to investigate changes in HDL metabolism in risky pregnancies and those complicated by preeclampsia. To achieve this goal, we analyze the concentrations of cholesterol synthesis and absorption markers within HDL particles, the activities of principal modulators of HDL cholesterol’s content, and the levels of major functional proteins associated with HDL in mid and late pregnancy.

## 2. Results

Although 114 women with risky pregnancies were initially included in the study, 90 were followed throughout the entire pregnancy. Twenty-four women were excluded from the study; 16 women dropped out of the survey, four were ruled out due to miscarriage, and four were excluded due to the appearance of fetal anomalies. Out of 90 women, 20 (22.2%) showed clinical signs of preeclampsia by the end of gestation. Ten women had preeclampsia as the only complication, whereas 10 had preeclampsia and secondary pregnancy complications; four had IUGR, and six had gestational diabetes. All women were diagnosed with late-onset preeclampsia, with 16 giving birth between the 34th and 37th weeks of gestation and four giving birth after week 37. Although 70 (77.8%) women did not develop preeclampsia despite being at risk, some had other pregnancy complications. Out of those 70 women, nine had only pregnancy hypertension, five of them had IUGR, four had gestational diabetes mellitus, two women had pregnancy hypertension and gestational diabetes, and two had IUGR and gestational diabetes mellitus, while one woman had three pregnancy complications—pregnancy hypertension, gestational diabetes, and IUGR. Additionally, one woman had diabetes type I, and three women had diabetes type II before pregnancy. Forty-seven women finished their pregnancies without developing complications.

For the study purpose, women were classified into two groups on the basis of the primary outcome—the risk group (RG), which did not develop preeclampsia, and the preeclampsia group (PG).

Basic clinical and lipid profile parameters in two test points in both groups are presented in Table 1. Although a statistically significant increase in mean arterial pressure (MAP) was observed in both groups, we found significantly higher MAP (*p* < 0.05) in PG compared with RG in the second and third trimesters. Women in PG had higher body weight, measured as body mass index (BMI), even before gestation (median: 23.7 kg/m^2^; interquartile range: 22.7–24.7 kg/m^2^ in RG vs. median: 26.2 kg/m^2^; interquartile range: 24.0–28.6 kg/m^2^ in PG, *p* < 0.05). PG was characterized by a higher BMI during the whole follow-up period (*p* < 0.05, in both test points) (Table 1). Additional comparisons of basic clinical and lipid profile parameters in the first and second trimesters can be found in the Appendix A. None of the women included in the study reported regular alcohol consumption. Nine women from PG (45.0%) and 21 (30.0%) women from RG reported positive smoking status before pregnancy. However, all women stated that they quit smoking during pregnancy. There was no significant difference in the smoking status between the two study groups before pregnancy (Pearson’s χ^2^ = 1.575, *p* = 0.209).

Even though an increase in total cholesterol concentration between second and third trimesters was observed in both groups, we did not find significant changes in LDL and HDL cholesterol concentrations between trimesters in PG. In both study groups, a significant increase in triglyceride concentration between the second and third trimesters was observed (*p* < 0.001). The concentration of HDL cholesterol was significantly lower in the second trimester in PG compared to RG (*p* < 0.05). Women in PG had significantly higher triglycerides levels in the second trimester (*p* < 0.05), while triglycerides concentrations in the third trimester were also higher in the PG but of borderline significance (*p* = 0.051) (Table 1).

A significant drop in the concentration of HDL cholesterol between the second and third trimesters in RG was accompanied by changes in the level of cholesterol synthesis and absorption markers in this lipoprotein fraction. We observed a drop in 7-dehydrocholesterol_HDL_ (*p* < 0.05), a cholesterol synthesis marker, as well as declines in campesterol_HDL_ (*p* < 0.001) and β-sitosterol_HDL_ (*p* < 0.001), i.e., cholesterol absorption markers, in RG in the third trimester (Table 2). No significant changes in the concentrations of NCS in the HDL fraction were observed in PG (Table 2). Additionally, campesterol_HDL_ and β-sitosterol_HDL_ levels were significantly lower in PG compared to RG in the second trimester. There were no significant differences in NCS_HDL_ between study groups in the third trimester (Table 2).

A significant decrease in the activity of LCAT (*p* < 0.05) and CETP (*p* < 0.05) between the second and third trimesters was observed in RG (Table 3). Additionally, we noticed a positive correlation between HDL cholesterol and LCAT activity in the second trimester in the same group (ρ = 0.382, *p* < 0.05). This correlation was absent in the third trimester (ρ = 0.149, *p* = 0.230). On the other hand, no significant correlations were observed between HDL cholesterol and LCAT activity in any of the testing points in PG (ρ = 0.454, *p* = 0.067, in the second trimester; ρ = 0.198, *p* = 0.430 in the third trimester). In PG, there were no significant changes in the activity of LCAT. Although there were no significant differences in CETP activity in PG, a trend toward lower activity was observed in the third trimester (Table 3). There were no statistically significant differences between the two study groups in LCAT and CETP activities (Table 3).

While a significant decrease in PON1 activity was observed in RG (*p* < 0.001) (Table 3), in PG, in the third trimester, a slightly higher PON1 activity was observed, but the change was not statistically significant (*p* = 0.313). PON1 activity was higher in the third trimester in PG than in RG (*p* < 0.05). Changes in the mass concentrations of apoA-I across trimesters accompanied the changes in the concentration of HDL cholesterol in both study groups. ApoA-I concentration significantly decreased between the 2nd and 3rd trimesters in RG, while there was no change in apoA-I levels in PG (Table 3). Additionally, a significant positive correlation between the PON1 activity and apoA-I concentration was seen in the second trimester in RG (ρ = 0.339, *p* < 0.05), while this was not found in PG (ρ = 0.058, *p* = 0.818).

## 3. Discussion

Physiological pregnancy is characterized by significantly, but transiently, altered lipoprotein metabolism. These changes typically occur to meet maternal and fetal needs and are restored after delivery [8]. Noticeably, preeclampsia is generally associated with even more striking changes in lipid homeostasis [9]. Indeed, significant changes in lipid homeostasis during pregnancy were also observed in women in our study. Pregnancy was accompanied by a longitudinal increase in the concentration of the basic lipid profile parameters in both study groups (Table 1). Apart from a notably higher concentration of triglycerides in PG, the only significant difference in lipid profile parameters between the two study groups was lower HDL cholesterol concentration in PG in the second trimester (Table 1). Previous studies have shown that women with physiological pregnancy experience increased HDL cholesterol concentration in the second trimester. At the same time, there is usually no such boost in women with pregnancy complications such as preeclampsia [9,21]. Low concentrations of HDL cholesterol are traditionally associated with the risk of developing atherosclerosis, cardiovascular disease, and diabetes [16].

The role and importance of HDL in pregnancy have been seldom investigated and, therefore, not sufficiently clarified. These particles could affect the sterol balance in the trophoblasts [22,23]. Videlicet, maternal HDL, could be an important source of cholesterol for the placenta during early pregnancy [22], and vice versa; HDL particles seem to mediate the cholesterol efflux from the trophoblast in mid and late gestation [23]. HDL particles carry a combination of anti-inflammatory proteins, including antioxidative enzymes, protease inhibitors, and proteins that affect the complement cascade. Furthermore, HDL affects immune cell activity by regulating sterol balance [22]. Keeping the endothelium intact and providing a sufficient maternal blood supply to the fetus is critical for preserving the pregnancy. HDL seems to play a major role in these processes via various functions, including stimulation of NO production [22]. We could assume that the decreased concentration of HDL cholesterol in the second trimester in PG is at least partially a consequence of the impaired capacity of HDL particles to take up cholesterol from peripheral tissues. As alluded to before, this might result in lower cholesterol efflux and sterol imbalance at the level of trophoblasts and immune cells prior to late-onset preeclampsia development.

Pregnancy is characterized by significant changes in the structure of HDL particles compared to nonpregnant women [11]. Previous studies have shown that the composition and size of HDL particles in women with preeclampsia are changed compared to physiological pregnancy [24]. Nevertheless, none of these studies examined HDL particles’ cholesterol metabolic profile.

We further analyzed the quantitative metabolic signature of cholesterol in the HDL plasma fraction. Our previous total plasma NCS analysis results suggested an imbalance in cholesterol homeostasis in women prone to preeclampsia development [21]. In previous studies, preeclampsia was associated with simultaneous increases in cholesterol synthesis and absorption [21,25]. However, to our knowledge, the content of NCS in the HDL fraction in women with preeclampsia and women with risky pregnancies has not been investigated so far. Detailed profiling of NCS performed in this study provided further insight into HDL particle structure. Lower concentrations of HDL cholesterol in women with preeclampsia in the second trimester were simultaneously followed by lower levels of cholesterol absorption markers (Table 2). On the contrary, we found no significant differences in endogenous cholesterol precursors (Table 2). Higher phytosterol levels are generally associated with lower cardiovascular disease risk [14]. Dietary cholesterol is absorbed by two independent pathways at the level of enterocytes: one including chylomicrons and the other involving an ATP-binding cassette subfamily A member 1 (ABCA1)/ApoA-I interaction and uptake by HDL particles (HDL pathway) [26]. It is considered that intestinal absorption of phytosterols occurs through a similar process of apoA-I lipidation [27]. In line with this, it has been proposed that phytosterol absorption via the HDL pathway could be used as a marker of intestinal ABCA1/ApoA-I activity [27]. Investigation in mice revealed that the intestinal HDL pathway could account for up to 30% of plasma HDL levels [27]. Thus, we could hypothesize that, in women prone to late-onset preeclampsia development, the capacity for synthesis of nascent HDL particles at the level of enterocytes is reduced. This assumption addresses new perspectives in therapeutic and preventive approaches for preeclampsia.

Additionally, altered lipid composition and the relationship between cholesterol synthesis and absorption markers (Table 1 and Table 2) were followed by different apoA-I content and PON1 activity (Table 3). Although PON1 is best known for its antioxidant properties, some recent research shed a completely different light on this enzyme and its role in regulating cholesterol homeostasis. ApoA-I is recognized as the carrier of PON1 on the HDL particle; in return, PON1 increases the apoA-I stability [18]. Furthermore, studies in mice have shown that increased PON1 expression results in improved HDL capacity for cholesterol efflux and, thus, increased reverse cholesterol transport [19,20]. It is generally accepted that PON1 activity increases throughout pregnancy; however, results regarding PON1 activity in preeclampsia are rather controversial [28,29,30,31,32]. While some authors claimed that PON1 activity is reduced in preeclampsia and suggested that this contributes to the development of the disease [28,29,30], others demonstrated higher serum PON1 activity in preeclampsia when compared to uncomplicated pregnancies [31,32]. We observed higher PON1 activity in PG compared to RG (Table 3). Additionally, in RG, the lowering of HDL cholesterol concentration was accompanied by a decrease in HDL phytosterol content, apoA-I concentration, and PON1 activity. On the other hand, in PG, all these longitudinal changes were absent (Table 1, Table 2 and Table 3). Therefore, we could speculate that the observed drop in HDL cholesterol in RG resulted from decreased ABCA1/ApoA-I activity in the intestine and reduced reverse cholesterol transport between the second and third trimesters. Conversely, these processes in PG seemed to be unchanged during gestation, corresponding to the lower levels observed in PG in the third trimester during the whole follow-up. Furthermore, we proposed that higher activity of PON1 in PG in the third trimester might be a response to lower HDL concentration in the second trimester; that is, it might represent an intrinsic adaptive mechanism aimed to enhance the reverse cholesterol transport alongside improved major antioxidative function of PON1.

We investigated the changes in LCAT and CETP, enzymes involved in cholesterol metabolism, and pivotal proteins in HDL particle maturation [12]. Lower LCAT activity is commonly associated with low HDL cholesterol levels and an increased risk of atherosclerosis and cardiovascular disease development [12]. In our study, higher HDL cholesterol concentrations in the second trimester (Table 1) were followed by somewhat higher, although not statistically significantly different, LCAT activity in RG compared to PG (Table 3). As anticipated, the decrease in HDL cholesterol (Table 1) and apoA-I, one of the major regulators of LCAT [12], was followed by a significant decline in LCAT activity between the second and third trimesters in RG (Table 3). Additionally, as expected, higher LCAT activity was associated with a higher concentration of HDL cholesterol in these women, which was confirmed by the positive correlation between these two variables in the second trimester reported in our study. On the other hand, our results showed no significant change in LCAT activity during pregnancy in women with preeclampsia, i.e., there was an absence of a typical answer of LCAT to a rise in cholesterol levels between trimesters in PG (Table 3). To our knowledge, this study was the first to examine LCAT in women with preeclampsia, and these preliminary results should be further explored.

As pregnancy is accompanied by a significant increase in the concentration of triglycerides, a matching increase in CETP activity is usually observed in women with uncomplicated pregnancies [8]. Although changes in CETP activity have been described in detail in physiological pregnancy [8] and some maternal and fetal complications [15], there is a lack of research related to the examination of the activity of this enzyme, which is crucial for the remodeling of HDL particles in women with preeclampsia. Our results indicated that the decrease in HDL concentrations was followed by a drop in the CETP activity in RG (Table 3). Additionally, we observed increased triglyceride concentration in RG (Table 1). We could assume that, taken together, plasma triglycerides do not play a substantial role in structural changes of HDL particles in RG. More likely, lower HDL cholesterol levels resulted from different mechanisms—decreased ABCA1/ApoA-I system and LCAT activities. Although there was no significant change in CETP activity between the second and third trimesters in PG, a trend to lower activity was observed in the third trimester (Table 3). This further confirms that factors other than CETP and triglyceride enrichment seem to be more critical for HDL remodeling in late-onset preeclampsia.

Several limitations should be mentioned. The number of subjects in PG was relatively small. The absence of a completely healthy cohort could have biased the results, as the differences were assessed in women with underlying health conditions or risk factors. These two major limitations should be addressed in future research involving more participants and a group of women with a healthy pregnancy and no risk factors.

In conclusion, it seems that the development of late-onset preeclampsia is preceded by the appearance of lower HDL cholesterol levels in the second trimester. We show for the first time that the reduced concentration of HDL cholesterol in these women at this point is accompanied by lower levels of cholesterol absorption markers (campesterol and β-sitosterol) in the HDL fraction. Reduced capacity for intestinal HDL synthesis, decreased LCAT activity, and impaired capacity for HDL-mediated cholesterol efflux could be the contributing mechanisms resulting in lower HDL cholesterol levels prior to late-onset preeclampsia, while CETP involvement seems to be of lesser importance.

## 4. Materials and Methods

### 4.1. Study Design

In this retrospective cohort study with sampling at three (for basic lipid profile parameters) or two points (for NCSs and HDL functionality markers) in longitudinal follow-up, we monitored 90 pregnant women longitudinally throughout gestation. Women were included in the study at their first antenatal checkup at the Obstetrics and Gynecology Clinic “Narodni Front” (Belgrade, Serbia). Ethical approval was obtained from the Ethics Committee of Gynecology and Obstetrics Clinic “Narodni front”, No. 05006-2020-10738; the Ethics Commission of the Faculty of Medicine, University of Belgrade, NUMBER: 1322/VII-27; and the Ethical Committee for Biomedical Research of the Faculty of Pharmacy, University of Belgrade, No. 1156/2. All subjects signed the informed written consent. During sampling and all experimental procedures, requirements set by institutional policies, national regulations, and ethical principles established by the Declaration of Helsinki were met. This study complies with the guidelines for human studies and was conducted in accordance with the World Medical Association Declaration of Helsinki.

### 4.2. Patients

Women with suspected risky pregnancies underwent early screening for preeclampsia. The screening was performed by the end of the first trimester, between 11 weeks + 0 days and 13 weeks + 6 days of gestation, ensuring that women with risky pregnancies were intensively monitored from the beginning of pregnancy and the frequency of pregnancy complications reduced to a minimum.

Study participants were recruited on the basis of insufficient flow through the uterine artery, measured by a pulse color Doppler, or the existing a priori risk for preeclampsia development, as evaluated by the guidelines of the National Institute for Health and Care Excellence (NICE) [33]. A woman had a high risk of developing preeclampsia if she had one high-risk factor or at least two moderate risk factors. According to the NICE criteria, high-risk factors for preeclampsia development are the presence of hypertension in a previous pregnancy, chronic hypertension, chronic kidney disease, type I or type II diabetes, and autoimmune disease. Moderate risk factors include first pregnancy, high maternal age (≥40 years), pregnancy interval of more than 10 years, BMI of 30 kg/m^2^ or more at the first visit, and family history of preeclampsia [33]. Although the primary outcome was preeclampsia, we also monitored the development of other pregnancy complications: pregnancy hypertension, IUGR, and gestational diabetes mellitus. The preeclampsia was diagnosed on the basis of new-onset hypertension observed after the 20th week of gestation followed by subsequent proteinuria.

### 4.3. Sample Collection

All women were fully monitored during the entire pregnancy. Blood was sampled at two points—in the second trimester (22–25 weeks of gestation), before the appearance of clinical signs of preeclampsia, and in the third trimester (35–38 weeks of gestation), when clinical signs of late-onset preeclampsia manifested, for the analysis of NCSs and HDL functionality markers. In addition, for the analysis of the basic parameters of the lipid profile, blood was also sampled at the beginning of pregnancy, i.e., in the first trimester (12–14 weeks of gestation). Venous blood samples were obtained after an overnight fast (≥12 h) and collected into serum and plasma sample tubes (Becton, Dickinson and Company, Franklin Lakes, NJ, USA). Samples were centrifuged at 1500× *g* for 10 min to obtain serum or plasma, and aliquots were stored at −80 °C until analyzed.

### 4.4. Methods

The arterial blood pressure was measured by a standardized procedure and MAP was calculated according to the following formula [34]:(1)$$\mathrm{MAP}=\frac{\mathrm{SBP}+2\;\times \;\mathrm{DBP}}{3}.$$

The height and body weight were measured to assess BMI—calculated according to the following formula:(2)$$\mathrm{BMI}=\frac{\mathrm{weight}\left(\mathrm{kg}\right)}{{\mathrm{height}}^{2}(m^{2})}.$$

Serum total cholesterol, triglyceride, and HDL cholesterol levels were measured using commercial tests (Beckman Coulter, Brea, CA, USA), while LDL cholesterol levels were calculated according to the Friedewald equation [35]. The concentration of ApoA-I was determined using a commercial kit (Beckman Coulter, Brea, CA, USA) on an automated biochemical analyzer Beckman AU 480 (Beckman Coulter, Brea, CA, USA).

Serum cholesterol precursors (desmosterol, 7-dehydrocholesterol, and lathosterol) and cholesterol absorption markers (campesterol and β-sitosterol), i.e., NCS_HDL_, were quantified by liquid chromatography–tandem mass spectrometry (LC-MS/MS), as previously reported [36]. Desmosterol, 7-dehydrocholesterol, lathosterol, campesterol, and β-sitosterol, as well as the deuterated internal standard—d6-cholesterol, were quantified using HPLC-grade analytical standards (Sepelco, Bellefonte, PA, USA) purchased from Sigma-Aldrich (St. Louis, MO, USA). KOH was acquired from POCH (Center Valley, PA, USA), and ethanol, methanol, n-hexane, and acetonitrile (HPLC grade) were obtained from Fisher (Pittsburgh, PA, USA). NCS_HDL_ were determined in ApoB-depleted serum after 200 μL of serum was added to 500 μL of ApoB-precipitation reagent containing phosphotungstate (0.4 mmol/L) and magnesium chloride (20 mmol/L) (BioSystems, Barcelona, Spain). After thorough vortexing, with 10 min incubation at room temperature, the mixture was centrifuged at 6000 rpm for 10 min. In the following step, 650 µL of supernatant containing HDL fraction of serum was removed and transferred into a glass conical tube previously coated with internal standard (50 µL of d6-cholesterol, 1 mg/mL). After short vortexing, 1 mL of 2% KOH in ethanol was added and vortexed for 15 s to provide protein precipitation. The alkaline hydrolysis of sterol esters was accomplished by incubating the mixture for 30 min at 45 °C. After cooling down to room temperature, 500 µL of HPLC-grade water was added to each sample. Subsequently, 2 mL of n-hexane was added, and the mixture was vigorously vortexed for 30 s. After centrifugation for 5 min at 1500× *g*, the upper layer was vigilantly transferred into another clean glass tube. The hexane extraction process was repeated three times in total, and all organic extracts were collected together. Excess KOH was washed off by adding 4 mL of HPLC-grade water in the organic extracts, followed by additional centrifugation for 5 min at 1500× *g*. The extract was then carefully collected, transferred into a clean glass tube, dried under the gentle stream of nitrogen, and reconstituted in 20 µL of HPLC-grade methanol. Finally, 10 µL of methanol extract was injected into the column. LC analysis was performed with an Agilent 1290 liquid chromatograph equipped with Poroshell 120 EC column (2.5 µm, 4.6 × 150 mm) (Agilent Technologies, Santa Clara, CA, USA) [36]. Chromatographic conditions included isocratic elution with a constant mobile phase flow of 0.6 mL/min and column temperature of 30 °C for a total of 45 min. Acetonitrile, methanol, and water with 0.1% formic acid (80:18:2, *v*/*v*) comprised the mobile phase. Quantification was performed using multiple-reaction monitoring on a triple-quad mass spectrometer Agilent 6420 with APCI as an ion source (Agilent Technologies, Santa Clara, CA, USA) [36]. The *m*/*z* transitions for each analyte were previously presented in detail [36]. The mass spectrometer was operated in positive APCI mode. The source conditions were as follows: gas temperature of 325 °C, vaporizer temperature of 250 °C, gas flow of 5 L/min, nebulizer pressure of 30 psi, positive capillary voltage of 2000 V, positive corona current of 4 µA, and positive charging of 2000 V [36]. The intra- and inter-run variabilities for all NCS_HDL_ were in the range from 2.5% to 13.6%. The recoveries for the same analytes were 85.3–95.8%, as reported previously [36].

LCAT and CETP activities were determined as described by Asztalos et al. [37]. LCAT activity was determined by measuring plasma cholesterol esterification rates as the difference in the concentration of FC before and after in vitro incubation of plasma for 2 h at 37 °C since the decrease in FC is equal to the increase in plasma CE [37]:(3)LCAT activity=initial plasma FC −final plasma FC.

To determine CETP activity, the concentrations of FC and CE in the HDL fraction, i.e., apoB-depleted plasma, were determined before and after 2 h incubation at 37 °C [37]. ApoB-depleted plasma was obtained as described above using BioSystems reagents (Barcelona, Spain). The degree of CE transport mediated by CETP from HDL to LDL and VLDL particles represents the difference between decrease in FC in plasma and the increase in CE in the HDL fraction, as a function of time [37]:$$\mathrm{CETP}\;\mathrm{activity}=\left[\mathrm{initial}\;\mathrm{plasma}\;\mathrm{FC}-\mathrm{final}\;\mathrm{plasma}\;\mathrm{FC}\right]\;-\left[\left(\mathrm{final}\;\mathrm{HDL}\;\mathrm{cholesterol}-\mathrm{final}\;\mathrm{HDL}\;\mathrm{FC}\right)-\left(\mathrm{initial}\;\mathrm{HDL}\;\mathrm{cholesterol}-\mathrm{initial}\;\mathrm{HDL}\;\mathrm{FC}\right)\right].$$

Reagents for FC and total cholesterol determination were purchased from Bioanalytica (Belgrade, Serbia). All measurements were performed on a Mindray BS200E analyzer (Mindray Bio-Medical Electronics Co., Ltd., Shenzhen, China) with intra-assay and inter-assay coefficients of variation <5%.

Serum paraoxonase 1 (PON1) activity was measured kinetically using paraoxon (Chem Service, West Chester, PA, USA) as a substrate, as previously described by Richter and Furlong [38]. The determination of the paraoxonase activity is based on the action of PON1 enzyme from the serum on the paraoxon substrate, whereby conversion of paraoxon to p-nitrophenol occurs. The velocity of the change is monitored kinetically at 405 nm, which is the characteristic absorption maximum for p-nitrophenol. PON1 activity was determined at 25 °C and at pH = 8.5 using 50 mmol/L TRIS-HCl buffer in the presence of NaCl [38]. The measurement was performed on a biochemical analyzer Ilab 300+ (Instrumentation Laboratory, Milan, Italy). The intra-assay coefficient of variation for PON1 activity was 5.4%, while the inter-assay coefficient of variation for the same method was 7.7%, as reported previously [39].

### 4.5. Statistical Analysis

Data distribution was tested using Kolmogorov–Smirnov and Shapiro–Wilk tests. The data were shown as the arithmetic mean and standard deviation for normally distributed variables, as the geometric mean and the 95th confidence intervals derived from log-normal values, and as the median and interquartile range for parameters not normally distributed even after logarithmic transformation. We used a paired-samples *t*-test or its nonparametric analog, Wilcoxon test, to assess the differences in the data as a function of time. The analysis of covariance (ANCOVA) or nonparametric ANCOVA (Quade method) was used to compare the means or medians between the two study groups, RG and PG, thereby regarding diabetic status as a covariate. We used Spearman correlation analysis to evaluate the correlation between parameters of interest. The calculated power of the study was higher than 0.8.

Statistical tests were considered significant at the 0.05 probability level. Statistical analyses were performed using PASW Statistics 18 (IBM, Armonk, NY, USA).

## Figures and Tables

**Table 1 ijms-24-11357-t001:** Changes in lipid profile parameters in RG and PG.

	2nd Trimester		3rd Trimester		p_1_	p_2_
	RG (N = 70)	PG (N = 20)	RG (N = 70)	PG (N = 20)		
WG	23.3 ± 0.83	23.4 ± 0.99	36.9 ± 0.89	36.6 ± 1.02		
Age, years	32.0 ± 5.56	33.1 ± 4.26				
MAP, mm Hg	82.5 ± 11.51	89.8 ± 9.64 ^dg^	88.2 ± 9.23	98.2 ± 11.88 ^df^	<0.001	<0.05
BMI ^b^, kg/m^2^	25.4 (23.6–30.0)	28.4 ^eg^ (26.4–32.7)	28.1 (25.5–32.4)	31.2 ^eg^ (28.6–36.1)	<0.001 ^c^	<0.001 ^c^
Weight gain ^b^, kg	5.40 (3.72–7.02)	4.70 (2.82–6.75)	3.80 (2.00–5.00)	4.30 (1.25–6.00)	<0.001 ^c^	0.232 ^c^
Weight gain ^b^, %	7.00 (5.00–9.75)	5.50 (4.00–8.75)	5.00 (3.00–6.00)	4.65 (1.25–7.00)	<0.001 ^c^	0.075 ^c^
TC, mmol/L	6.81 ± 1.365	6.61 ± 1.236	7.49 ± 1.611	7.17 ± 1.506	<0.001	<0.05
HDL-C, mmol/L	2.11 ± 0.386	1.88 ± 0.363 ^dg^	1.97 ± 0.520	1.91 ± 0.310	<0.05	0.738
LDL-C, mmol/L	3.81 ± 1.148	3.61 ± 1.104	4.16 ± 1.273	3.51 ± 1.655	<0.05	0.953
TG ^a^, mmol/L	1.86 (1.73–2.00)	2.33 ^dg^ (2.00–2.71)	2.93 (2.72–3.16)	3.52 (3.07–4.04)	<0.001	<0.001

RG—risk group; PG—preeclampsia group; WG—week of gestation; BMI—body mass index; MAP—mean arterial pressure; TC—total cholesterol; HDL-C—high density lipoprotein cholesterol; LDL-C—low density lipoprotein cholesterol; TG—triglycerides. Data are shown as the mean ± standard deviation. ^a^ Geometric mean (95th CI); ^b^ median (interquartile range). p_1_—Paired-samples *t*-test for risk group; p_2_—Paired-samples *t*-test for preeclampsia group. p_1_
^c^—Wilcoxon test for risk group; p_2_
^c^—Wilcoxon test for preeclampsia group. Significantly different from the risk group: ^d^ analysis of covariance (ANCOVA); ^e^ nonparametric ANCOVA (Quade method). ^f^ *p* < 0.001; ^g^ *p* < 0.05.

**Table 2 ijms-24-11357-t002:** Changes in NCS_HDL_ concentrations in RG and PG.

	2nd Trimester		3rd Trimester		p_1_	p_2_
	RG (N = 70)	PG (N = 20)	RG (N = 70)	PG (N = 20)		
WG	23.3 ± 0.83	23.4 ± 0.99	36.9 ± 0.89	36.6 ± 1.02		
Desmosterol_HDL_, μmol/L	0.19 (0.13–0.24)	0.17 (0.13–0.23)	0.23 (0.15–0.32)	0.21 (0.12–0.26)	0.139	0.095
7-DHC_HDL_, μmol/L	0.36 (0.28–0.51)	0.40 (0.31–0.47)	0.33 (0.25–0.44)	0.37 (0.25–0.57)	<0.05	0.904
Lathosterol_HDL_, μmol/L	1.24 (0.73–1.87)	1.22 (0.64–1.89)	1.16 (0.70–2.01)	1.24 (0.51–2.11)	0.677	0.931
Campesterol_HDL_, μmol/L	0.77 (0.55–1.11)	0.44 ^dg^ (0.24–0.66)	0.62 (0.36–0.81)	0.53 (0.26–0.69)	<0.001	0.355
β-Sitosterol_HDL_, μmol/L	2.88 (2.11–3.96)	2.26 ^dg^ (1.54–3.24)	2.40 (1.80–3.36)	2.32 (1.83–3.97)	<0.001	0.409

RG—risk group; PG—preeclampsia group; NCS_HDL_—non-cholesterol sterols in high-density lipoprotein (HDL) fraction; WG—week of gestation; 7-DHC—7-dehydrocholesterol. Data are shown as the median (interquartile range). p_1_—Wilcoxon test for risk group; p_2_—Wilcoxon test for preeclampsia group. Significantly different from the risk group: ^d^ nonparametric analysis of covariance (ANCOVA) (Quade method). ^g^ *p* < 0.05.

**Table 3 ijms-24-11357-t003:** Changes in HDL functionality markers in RG and PG.

	2nd Trimester		3rd Trimester		p_1_	p_2_
	RG (N = 70)	PG (N = 20)	RG (N = 70)	PG (N = 20)		
WG	23.3 ± 0.83	23.4 ± 0.99	36.9 ± 0.89	36.6 ± 1.02		
LCAT ^b^, μmol/L/h	113.2 (71.0–152.5)	94.2 (65.1–141.3)	77.0 (54.7–106.9)	97.0 (79.0–111.7)	<0.05 ^c^	0.463 ^c^
CETP ^b^, μmol/L/h	65.2 (24.2–127.2)	62.0 (11.7–106.1)	34.7 (13.2–63.8)	32.9 (18.5–72.0)	<0.05 ^c^	0.133 ^c^
PON1 ^b^, U/L	351.0 (263.0–863.0)	697.5 (379.0–1024.7)	347.0 (235.5–699.0)	771.0 ^eg^ (357.7–1162.0)	<0.001 ^c^	0.313 ^c^
ApoA-I ^a^, g/L	2.32 ± 0.347	22.29 ± 0.445	2.23 ± 0.407	2.34 ± 0.425	<0.05	0.501

RG—risk group; PG—preeclampsia group; WG—week of gestation; LCAT—lecithin—cholesterol acyltransferase; CETP—cholesterol ester transfer protein; PON1—paraoxonase 1; ApoA-I—apolipoprotein A-I. Data are shown as the mean ± standard deviation. ^a^ Geometric mean (95th CI); ^b^ median (interquartile range). p_1_—Paired-samples *t*-test for risk group; p_2_—Paired-samples *t*-test for preeclampsia group. p_1_
^c^—Wilcoxon test for risk group; p_2_
^c^— Wilcoxon test for preeclampsia group. Significantly different from the risk group: ^e^ nonparametric ANCOVA (Quade method). ^g^ *p* < 0.05.

## Data Availability

The data presented in this study are available on request from the corresponding author. The data are not publicly available due to privacy and ethics considerations.

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
