# Peer review of "Cholesterol Metabolic Profiling of HDL in Women with Late-Onset Preeclampsia"

_ijms, 2023, doi:10.3390/ijms241411357_

Round 1
Reviewer 1 Report
the authors extend a mechanistic insight into parameters in lipid metabolism in preeclampsia vs those presumed to be at risk because of screening with Doppler evaluation of uterine blood flow early in pregnancy
heretofore not studied they put forth mechanistic inference that targets their analysis while methods are heavily displayed the absences of indices of validity and reproducibility and observer variation are concerning
they describe weaknesses that are essential this is a longitudinal study with sampling at 2-time points not strictly cross-sectional as they observe outcomes defined as PG or "RG" (so apparently all patients are resampled within their diagnosis. There is no mention of diabetes status alcohol use or smoking all of which can affect lipid metabolism They do not fully adjust in their analysis that being said their desire to enhance understanding of the aberrant lipid metabolism should be applauded they are careful to separate late-onset preeclampsia in their introduction and they clarify that is the variety studied here
Author Response
Reply to the Review Report (Reviewer 1)
A point-by-point reply follows:
Reviewers’ comments:
Reviewer #1 comment 1: the authors extend a mechanistic insight into parameters in lipid metabolism in preeclampsia vs those presumed to be at risk because of screening with Doppler evaluation of uterine blood flow early in pregnancy
Answer: Thank you very much for pointing this out. Our main aim was to obtain deeper insight into the changes in the cholesterol metabolism regulation and specific mechanisms of cholesterol homeostasis in women with late-onset preeclampsia. We believe our research provided a clearer picture of the significance of the changes in lipid metabolism in high risk pregnancies and pregnancies followed by preeclampsia. One of the inclusion criteria for the study participants were the insufficient flow through the uterine arteries, measured by a pulse color Doppler, or the existing a priori risk for preeclampsia development, as recommended by the guidelines of the National Institute for Health and Care Excellence (NICE).
Reference: National Institute of Health and Excellence. Quality standard [QS35]: Hypertension in pregnancy. National Institute of Health and Excellence: Manchester, United Kingdom, 2013.
Reviewer #1 comment 2: heretofore not studied they put forth mechanistic inference that targets their analysis while methods are heavily displayed the absences of indices of validity and reproducibility and observer variation are concerning
Answer: Thank you for pointing this out. In the Materials and Methods section, we have added information about appropriate coefficients of variation for all non-commercial methods, as follows:
- Method for NCSHDL determination (Page 10 in the revised version):
“The intra- and inter-run variabilities for all NCSHDL were in the range from 2.5% to 13.6%. The recoveries for the same analytes were 85.3-95.8, as reported previously [40].”
- Method for LCAT and CETP determination (Page 11 in the revised version):
“All measurements were performed on a Mindray BS200E analyzer (Mindray Bio-Medical Electronics Co., Ltd., Shenzhen, China) with intra-assay and inter-assay coefficients of variation <5%.”
- Method for PON1 activity measurement (Page 11 in the revised version):
“The intra-assay coefficient of variation for PON1 activity was 5.4%, while the inter-assay coefficient of variation for the same method was 7.7%, as reported previously [43].”
Reviewer #1 comment 3: they describe weaknesses that are essential this is a longitudinal study with sampling at 2-time points not strictly cross-sectional as they observe outcomes defined as PG or "RG" (so apparently all patients are resampled within their diagnosis.
Answer: Thank you very much for the comment.
Our study was designed as a longitudinal study, all pregnant women were monitored throughout the entire pregnancy (1st, 2nd, 3rd trimester and point before delivery). However, for the purpose of this manuscript, we concentrated only on the two points, to simplify the presentation of the observed data and results. Two points were targeted – the second trimester of pregnancy, before the appearance of clinical signs of preeclampsia, when important changes in the metabolism of HDL are expected (increase in HDL-C concentration in pregnancies without complications), and the late third trimester, when the diagnosis of late-onset preeclampsia was already set in the PG.
Reviewer #1 comment 4: There is no mention of diabetes status alcohol use or smoking all of which can affect lipid metabolism. They do not fully adjust in their analysis that being said their desire to enhance understanding of the aberrant lipid metabolism should be applauded.
Answer: Thank you very much for mentioning this.
Examining the effect of diabetes on lipid metabolism is beyond the scope of this study. According to the guidelines of the National Institute for Health and Care Excellence (NICE) presence of diabetes type I and type II are listed as high-risk factors for preeclampsia development. This was added in the Materials and Methods section (Page 10 in the revised manuscript version).
In this study group, one woman had diabetes type I, and three women had diabetes type II before pregnancy (Page 4 in the revised Results section):
“Additionally, one woman had diabetes type I, and three women had diabetes type II before pregnancy.”
Also, in the first version of the manuscript, we already stated that a part of our subjects developed gestational diabetes during the observational period.
“All women were diagnosed with late-onset preeclampsia, with 16 giving birth between the 34 and 37 weeks of gestation and four after week 37. Although 70 (77.8%) women did not develop preeclampsia despite being at risk, some had other pregnancy complications. Out of those 70 women, nine had only pregnancy hypertension, five of them had IUGR, four had gestational diabetes mellitus, two women had pregnancy hypertension and gestational diabetes, and two had IUGR and gestational diabetes mellitus while one woman had three pregnancy complications – pregnancy hypertension, gestational diabetes, and IUGR.”
Alcohol consumption and smoking status were investigated through questionnaires filled out by the respondents. None of the women included in the study reported regular alcohol consumption. Nine women from PG (45.0%) and 21 (30.0%) women from RG reported positive smoking status before pregnancy. However, all women stated that they quit smoking during pregnancy. As tested by Pearson’s Chi-Square test, there was no significant difference in the smoking status between the two study groups before pregnancy (P=0.209). Following your kind suggestion, these informations were added in the Results section (Page 4 in the revised manuscript version):
“None of the women included in the study reported regular alcohol consumption. Nine women from PG (45.0%) and 21 (30.0%) women from RG reported positive smoking status before pregnancy. However, all women stated that they quit smoking during pregnancy. There was no significant difference in the smoking status between the two study groups before pregnancy (Pearson’s Χ2=1.575, P=0.209).”
Reviewer #1 comment 5: they are careful to separate late-onset preeclampsia in their introduction and they clarify that is the variety studied here
Answer: Thank you very much. As emphasized in the manuscript, all women in our study were diagnosed with late-onset preeclampsia. We have made substantial changes in the Introduction section to clarify the differences between the two types of preeclampsia (Page 2 in the revised manuscript version):
„Both early- and late-onset preeclampsia are thought to develop in two stages. The first stage includes poor perfusion of the placenta and consequent placental dysfunction. In response to inadequate blood supply to the placenta and consequent hypoxia, oxidative stress develops and increased production of free radicals and inflammatory cytokines by the placenta emerges. These molecular mediators of oxidative stress and inflammation lead to the development of generalized maternal endothelial dysfunction in the second stage of the disease development [4,5]. However, the key difference between early- and late-onset preeclampsia lies in the cause and time of placental malperfusion and dysfunction [4]. In the early-onset syndrome, the first stage is the result of poor placentation due to the shallow remodeling of spiral arteries during the first half of pregnancy. Incomplete remodeling of spiral arteries underlies abnormal uteroplacental perfusion, leading to placental oxidative stress and causing the hypersecretion of inflammatory and antiangiogenic factors into the maternal circulation [4]. In women with late-onset preeclampsia, the maturating placenta outgrows the uterus capacity and the supporting maternal functions. In this case, the placenta also becomes under-perfused, with restricted intervillous perfusion, causing placental stress at a later gestational age [4,5]. Placental overgrowth seems to be particularly associated with maternal obesity and large placentas [4]. Hence, both pathways cause placental hypoperfusion and dysfunction in the first stage, leading to the clinically recognized maternal syndrome of preeclampsia in the second stage of disease development. Early-onset syndrome is associated with adverse maternal and neonatal outcomes, such as reduction in placental volume, intrauterine growth restriction (IUGR), abnormal uterine and umbilical artery Doppler evaluation, low birth weight, multiorgan dysfunction, and perinatal death [5,6]. On the other hand, the late-onset disorder is usually associated with a larger placental volume, normal uterine and umbilical artery Doppler evaluation, a normally grown baby with no signs of growth restriction, and significantly more favorable outcomes for the mother and fetus [5,6]. However, late-onset preeclampsia has a higher incidence (2.7% vs. 0.38%) [5,6].“
Reviewer 2 Report
The most used methodology ===without reference(LN/413-481--etc )
All corrections should be done before the start of publication process
There is no recommendations
There is no highlights (should be ) ???
Abstract needs more improvement and should contained background including aims , Methods , Results and conclusion
The authors used a huge a mounts of abbreviations , it will be better if you create a separate table for this
LN/14---adsorption markers---more details are needed
LN/22----longitudinal changes ----why ????
LN/27---add preeclampsia , NCShdl and biochemical indices to the keywords
LN/33----signs of organ dysfunction ----such what ???
Tabulate the different types of preeclampsia , with the characteristic features for each , causes and main complications and treatment associated with an update references
LN/38-39---explain in detailed manner
LN/40---inflammatory molecules---do you mean for example cytokines or what ????
LN/41---endothelial dysfunction ---what about the pathogenesis ???
LN/50/52/67----add references
Introduction is extremely long ---why ??? be more concise and summarize
LN/364----materials and methods should be cited after introduction and before results and discussion (reasonable )
LN/367---gestation longitudinally ---add reference
LN/379----add the total number of patients and classification of the groups have done according to whom ????
The most used methodology ===without reference(LN/413-481--etc )
There is no reference for the statistical analysis
Write as Table(1):----------------and Fig.(1):----------etc---apply for all
The authors did not mention any thing about all of the followings :-
a- The recorded clinical signs observed during running this work
b -The recorded mortalities percentages ---if there ---in tables
c-Score lesions --if there
LN/504-509---delete as it is previously mention at materials and methods
The most used cited references contained more than 6 authors ---why ??? should be 6 at the maximum plus etal with the last ones ---apply for all(ref---8,9,14,19,28,29,30----etc)
Some cited references need to be more update
As volume , issue , number and pages ---all are available ----so no need for the link(s)---apply for all
There is no plan for the study area
There is no charts or graphics
LN/526----why all capitals ???
Some journal names were written abbreviated , while others were not ---why ??? same style should be ---apply for all
LN/134---which is better retardation or restriction --and why ???
Early and late preeclampsia (34-37 weeks )---why and what are the prominent features or changes that occurred either to the mother or fetus during this time ????
LN/163-172----be more summarize and apply for all
LN/228-233----where is the discussion ???
LN/234-242---this is like introduction not a discussion
Discussion should be rewritten again and be based upon debating the obtained results with those of the previous investigators results

Okay
Author Response
Reviewer #2 comment 1: The most used methodology without reference (LN/413-481--etc )
Answer: Thank you very much for the comment.
Serum total cholesterol, triglyceride, and HDL cholesterol levels were measured by commercial tests (Beckman Coulter, Brea, California, USA), while LDL cholesterol levels were calculated according to the Friedewald equation. The concentration of ApoA-I was determined using a commercial kit (Beckman Coulter, Brea, California, USA) on an automated biochemical analyzer Beckman AU 480 (Beckman Coulter, Brea, California, USA). Since all of these parameters were measured using commercially available tests, no specific method is referenced, but we have listed the name of the manufacturer for each test.
The concentrations of NCSHDL were quantified by the liquid chromatography-tandem mass spectrometry (LC-MS/MS) method, as previously reported in Vladimirov et al. (Reference: Vladimirov, S.; Gojković, T.; Zeljković, A.; Jelić-Ivanović, Z.; Spasojević-Kalimanovska, V. Determination of non-cholesterol sterols in serum and HDL fraction by LC/MS-MS: Significance of matrix-related interferences. J Med Biochem 2020, 39(3), 299-308. doi: 10.2478/jomb-2019-0044).
LCAT and CETP activities were determined as described by Asztalos et al. (Reference: Asztalos, B.F.; Swarbrick, M.M.; Schaefer, E.J.; Dallal, G.E.; Horvath, K.V.; Ai, M.; Stanhope, K.L.; Austrheim-Smith, I.; Wolfe, B.M.; Ali, M.; Havel, P.J. Effects of weight loss, induced by gastric bypass surgery, on HDL remodeling in obese women. J Lipid Res 2010, 51(8), 2405-2412. doi: 10.1194/jlr.P900015).
Serum paraoxonase 1 (PON1) activity was measured kinetically using paraoxon as a substrate, as previously described by Richter and Furlong (Reference: Richter, R.J.; Furlong, C.E. Determination of paraoxonase (PON1) status requires more than genotyping. Pharmacogenetics 1999, 9(6), 745-753).
Reviewer #2 comment 2: There is no recommendations
Answer: Thank you very much for the comment. According to the propositions given in the Instructions for Authors for the International Journal of Molecular Sciences, it is not necessary to list recommendations as a separate segment. Below is an example of an original article that was recently published in the International Journal of Molecular Sciences:
Falquet, M.; Prezioso, C.; Ludvigsen, M.; Bruun, J.A.; Passerini, S.; Sveinbjørnsson, B.; Pietropaolo, V.; Moens, U. Regulation of Transcriptional Activity of Merkel Cell Polyomavirus Large T-Antigen by PKA-Mediated Phosphorylation. Int J Mol Sci 2023, 24(1), 895. doi: 10.3390/ijms24010895.
Reviewer #2 comment 3: There is no highlights (should be ) ???
Answer: We appreciate the suggestion. However, following the detailed instruction given in the Instructions for Authors for the International Journal of Molecular Sciences, we did not find the information that the highlights should be mentioned in a separate section. Once again, we give an example of an article that was recently published in the journal that could be used as a respectable example:
Falquet, M.; Prezioso, C.; Ludvigsen, M.; Bruun, J.A.; Passerini, S.; Sveinbjørnsson, B.; Pietropaolo, V.; Moens, U. Regulation of Transcriptional Activity of Merkel Cell Polyomavirus Large T-Antigen by PKA-Mediated Phosphorylation. Int J Mol Sci 2023, 24(1), 895. doi: 10.3390/ijms24010895.
Reviewer #2 comment 4: Abstract needs more improvement and should contained background including aims, Methods, Results and conclusion
Answer: Appreciating the criticism, we have supplemented the abstract with additional background information. Also, we tried to make the abstract more comprehensible by giving full terms for each abbreviation used in the abstract (Page 1 in the revised manuscript). According to the propositions defined by the Instructions for Authors for the International Journal of Molecular Sciences, the abstract should be a single paragraph and should follow the style of structured abstracts, but without headings, so the usual headings (Background, Aim, Methods, Results, and Conclusion) were left out in the revised manuscript version (Page 1 in the revised manuscript). Trying to, as much as possible, respect the propositions given by the Journal regarding the number of words, unfortunately, we did not find room for further improvement of the abstract.
Reviewer #2 comment 5: The authors used a huge a mounts of abbreviations, it will be better if you create a separate table for this
Answer: Thank you for this kind suggestion.
A list of abbreviations with their meanings can be found in the table below. In agreement with the Editorial Board, it could be included in the manuscript.
|
Abbreviation |
Meaning |
|
IUGR |
intrauterine growth restriction |
|
LDL |
low-density lipoprotein |
|
HDL |
high-density lipoprotein |
|
NCS |
non-cholesterol sterols |
|
NCSHDL |
non-cholesterol sterols in HDL fraction |
|
LCAT |
lecithin: cholesterol acyltransferase |
|
CE |
cholesterol ester |
|
CETP |
cholesterol ester transfer protein |
|
VLDL |
very-low-density lipoprotein |
|
ApoA-I |
apolipoprotein A-I |
|
PON1 |
paraoxonase 1 |
|
RG |
the risk group |
|
PG |
the preeclampsia group |
|
BMI |
body mass index |
|
ABCA1 |
ATP binding cassette subfamily A member 1 |
Reviewer #2 comment 6: LN/14---adsorption markers---more details are needed
Answer: Thank you very much for this comment. In this study, concentrations of two cholesterol absorption markers, i.e., phytosterols (plant sterols) were determined: campesterol and β-sitosterol. In the revised abstract form (Page 1 in the revised version) we mentioned the absorption markers which were determined:
“We will analyze cholesterol synthesis (cholesterol precursors: desmosterol, 7-dehydrocholesterol, and lathosterol) and absorption markers (phytosterols: campesterol and β-sitosterol) within HDL particles (NCSHDL), the activities of principal modulators of HDL cholesterol’s content and major HDL functional proteins levels in mid and late pregnancy.”
Reviewer #2 comment 7: LN/22----longitudinal changes ----why????
Answer: Thank you much for pointing this out. Our study was designed as a longitudinal study, and all pregnant women were monitored throughout the entire pregnancy (1st, 2nd, 3rd trimester and point before delivery). However, for the purpose of this manuscript, we have decided to concentrate only on two points, to simplify the presentation of the observed data and results. Two points were targeted – the second trimester of pregnancy, before the appearance of clinical signs of preeclampsia, when important changes in the metabolism of HDL are expected, and the late third trimester, when the diagnosis of late-onset preeclampsia was already set in the PG. In the mentioned line, we specifically aimed to emphasize that there were no significant changes in the concentration of HDL cholesterol, tested phytosterols, apolipoprotein A-I, and paraoxonase 1, lecithin: cholesterol acyl-transferase, and cholesterol ester transfer protein between the second and third trimester in PG.
Reviewer #2 comment 8: LN/27---add preeclampsia, NCShdl and biochemical indices to the keywords
Answer: We are grateful for your suggestion. Preeclampsia, NCSHDL, and biochemical indices were added to the keywords, while the word “pregnancy” was removed from the list of keywords in order to comply with the Journal’s rule that up to ten pertinent words should be added after the abstract.
Reviewer #2 comment 9: LN/33----signs of organ dysfunction ----such what???
Answer: Thank you very much. The definition of preeclampsia has been updated in the revised Introduction section (Page 1 in the revised version):
„Preeclampsia is one of the most severe pregnancy complications, with an estimated incidence of 4.6% although wide variations are recorded across regions [1]. It is defined as new-onset hypertension diagnosed after the 20th week of gestation with subsequent proteinuria or some other signs of end-organ dysfunction, such as thrombocytopenia, damage to liver function, development of renal failure, pulmonary edema or new-onset cerebral or visual disturbances [2].“
Reviewer #2 comment 10: Tabulate the different types of preeclampsia, with the characteristic features for each, causes and main complications and treatment associated with an update references
Answer: We appreciate the criticism. We certainly agree with the Reviewer’s comment, so we have provided a more detailed explanation with key characteristics for both types of preeclampsia in the revised Introduction section (Page 2 in the revised version). We have also provided the updated references, as requested.
„Both early- and late-onset preeclampsia are thought to develop in two stages. The first stage includes poor perfusion of the placenta and consequent placental dysfunction. In response to the inadequate blood supply to the placenta and consequent hypoxia, oxidative stress develops and increased production of free radicals and inflammatory cytokines by the placenta emerges. These molecular mediators of oxidative stress and inflammation lead to the development of generalized maternal endothelial dysfunction in the second stage of the disease development [4,5]. However, the key difference between early- and late-onset preeclampsia lies in the cause and time of placental malperfusion and dysfunction [4]. In the early-onset syndrome, the first stage is the result of poor placentation due to the shallow remodeling of spiral arteries during the first half of pregnancy. Incomplete remodeling of spiral arteries underlies abnormal uteroplacental perfusion, leading to placental oxidative stress and causing the hypersecretion of inflammatory and antiangiogenic factors into the maternal circulation [4]. In women with late-onset preeclampsia, the maturating placenta outgrows the uterus capacity and the supporting maternal functions. In this case, the placenta also becomes under-perfused, with restricted intervillous perfusion, causing placental stress at a later gestational age [4,5]. Placental overgrowth seems to be particularly associated with maternal obesity and large placentas [4]. Hence, both pathways cause placental hypoperfusion and dysfunction in the first stage, leading to the clinically recognized maternal syndrome of preeclampsia in the second stage of disease development. Early-onset syndrome is associated with adverse maternal and neonatal outcomes, such as reduction in placental volume, intrauterine growth restriction (IUGR), abnormal uterine and umbilical artery Doppler evaluation, low birth weight, multiorgan dysfunction, and perinatal death [5,6]. On the other hand, the late-onset disorder is usually associated with a larger placental volume, normal uterine and umbilical artery Doppler evaluation, a normally grown baby with no signs of growth restriction, and significantly more favorable outcomes for the mother and fetus [5,6]. However, late-onset preeclampsia has a higher incidence (2.7% vs. 0.38%) [5,6].“
Reviewer #2 comment 11: LN/38-39---explain in detailed manner
Answer: Thank you very much for the suggestion. Following the Reviewer’s recommendation, the pathophysiological mechanisms and key differences between the two types of preeclampsia were explained in a detailed manner in the revised Introduction section (Page 2 in the revised manuscript):
„Both early- and late-onset preeclampsia are thought to develop in two stages. The first stage includes poor perfusion of the placenta and consequent placental dysfunction. In response to the inadequate blood supply to the placenta and consequent hypoxia, oxidative stress develops and increased production of free radicals and inflammatory cytokines by the placenta emerges. These molecular mediators of oxidative stress and inflammation lead to the development of generalized maternal endothelial dysfunction in the second stage of the disease development [4,5]. However, the key difference between early- and late-onset preeclampsia lies in the cause and time of placental malperfusion and dysfunction [4]. In the early-onset syndrome, the first stage is the result of poor placentation due to the shallow remodeling of spiral arteries during the first half of pregnancy. Incomplete remodeling of spiral arteries underlies abnormal uteroplacental perfusion, leading to placental oxidative stress and causing the hypersecretion of inflammatory and antiangiogenic factors into the maternal circulation [4]. In women with late-onset preeclampsia, the maturating placenta outgrows the uterus capacity and the supporting maternal functions. In this case, the placenta also becomes under-perfused, with restricted intervillous perfusion, causing placental stress at a later gestational age [4,5]. Placental overgrowth seems to be particularly associated with maternal obesity and large placentas [4]. Hence, both pathways cause placental hypoperfusion and dysfunction in the first stage, leading to the clinically recognized maternal syndrome of preeclampsia in the second stage of disease development.“
Reviewer #2 comment 12: LN/40---inflammatory molecules---do you mean for example cytokines or what????
Answer: Thank you very much for pointing this out. This was corrected in the revised version of the manuscript (Page 2 in the revised Introduction section):
„In response to the inadequate blood supply to the placenta and consequent hypoxia, oxidative stress develops and increased production of free radicals and inflammatory cytokines by the placenta emerges.“
Reviewer #2 comment 13: LN/41---endothelial dysfunction ---what about the pathogenesis???
Answer: Thank you for the comment.
As explained in the Introduction section and responses to comments 11 and 12, endothelial dysfunction, i.e., damage to the endothelium in women with preeclampsia is mostly caused by the free radicals and inflammatory cytokines, produced and released by the malperfused and dysfunctional placenta (Page 2 in the revisedversion):
„Both early- and late-onset preeclampsia are thought to develop in two stages. The first stage includes poor perfusion of the placenta and consequent placental dysfunction. In response to the inadequate blood supply to the placenta and consequent hypoxia, oxidative stress develops and increased production of free radicals and inflammatory cytokines by the placenta emerges. These molecular mediators of oxidative stress and inflammation lead to the development of generalized maternal endothelial dysfunction in the second stage of the disease development [4,5].“
Reviewer #2 comment 14: LN/50/52/67----add references
Answer: Thanks for pointing this out.
Given that in a subsequent comment the reviewer expressed the opinion that the introduction was too long, in an attempt to shorten the introduction, lines 50 and 52 were deleted in the revised Introduction section. Appropriate reference fot line 67 has been added in the revised manuscript (Page 3 in the revised Introduction section).
Reference: Yang, Y.; Wang, Y.; Lv, Y.; Ding, H. Dissecting the roles of lipids in preeclampsia. Metabolites 2022, 12(7), 590. doi: 10.3390/metabo12070590.
Reviewer #2 comment 15: Introduction is extremely long ---why??? be more concise and summarize
Answer: Thank you for the suggestion.
The Introduction section has been rewritten and some parts have been deleted in an attempt to shorten the section (Pages 1-4 in the revised manuscript). Although the introductory part is still extensive, we believe that all segments are necessary for a proper interpretation of the topic and manuscript.
Reviewer #2 comment 16: LN/364----materials and methods should be cited after introduction and before results and discussion (reasonable)
Answer: We appreciate the criticism.
Although the section related to Materials and Methods is most often listed after the Introduction and before the Results section, the propositions of the International Journal of Molecular Sciences are different. We have used the Microsoft Word template given by the Journal when preparing the manuscript for submission. Below is an example of an original article that was recently published in the International Journal of Molecular Sciences:
Falquet, M.; Prezioso, C.; Ludvigsen, M.; Bruun, J.A.; Passerini, S.; Sveinbjørnsson, B.; Pietropaolo, V.; Moens, U. Regulation of Transcriptional Activity of Merkel Cell Polyomavirus Large T-Antigen by PKA-Mediated Phosphorylation. Int J Mol Sci 2023, 24(1), 895. doi: 10.3390/ijms24010895.
Reviewer #2 comment 17: LN/367---gestation longitudinally ---add reference
Answer: Thank you very much for the comment.
Our study was designed as a longitudinal study, and all pregnant women were monitored throughout the pregnancy. However, for the purpose of this manuscript, we have decided to concentrate only on two points, to simplify the presentation of the observed data and results. Two points were targeted – the second trimester of pregnancy, before the appearance of clinical signs of preeclampsia, when significant changes in the metabolism of HDL are expected, and the late third trimester, when the diagnosis of late-onset preeclampsia was already set in the PG.
Line 367 relates to our assertion about the design of the study, which is part of the study conceptualization, and is contributed by Aleksandra Stefanović and Željko Miković, investigators included in the study.
Reviewer #2 comment 18: LN/379----add the total number of patients and classification of the groups have done according to whom????
Answer: Thank you for pointing this out.
According to the Journal's propositions, the section with Materials and Methods is listed at the end of the manuscript, so we felt that the description of the patients included in the study had to precede the presentation of the results so that the results of the study would be understandable to the reader. The classification of women into two groups, the risk group (RG), which did not develop preeclampsia, and the preeclampsia group (PG), was done based on the primary outcome. The diagnosis of preeclampsia was set following the American College of Obstetricians and Gynecologists (ACOG) criteria (Reference: The American College of Obstetricians and Gynecologists. ACOG Practice Bulletin No. 202: Gestational Hypertension and Preeclampsia. Obstet Gynecol 2019, 133(1), e1 25. doi: 10.1097/AOG.0000000000003018).
The requested data can be found in the Results section (Page 4 in the revised manuscript):
“Although 114 women with risky pregnancies were initially included in the study, 90 were followed throughout the entire pregnancy. Twenty-four women were excluded from the study – 16 women dropped out of the survey, four were ruled out due to miscarriage, and four due to the appearance of fetal anomalies. Out of 90 women, 20 (22.2%) showed clinical signs of preeclampsia by the end of gestation. Ten women had preeclampsia as the only complication, ten had preeclampsia and secondary pregnancy complications - four had intrauterine growth restriction (IUGR), and six had gestational diabetes. All women were diagnosed with late-onset preeclampsia, with 16 giving birth between the 34 and 37 weeks of gestation and four after week 37. Although 70 (77.8%) women did not develop preeclampsia despite being at risk, some had other pregnancy complications. Out of those 70 women, nine had only pregnancy hypertension, five of them had IUGR, four had gestational diabetes mellitus, two women had pregnancy hypertension and gestational diabetes, and two had IUGR and gestational diabetes mellitus while one woman had three pregnancy complications – pregnancy hypertension, gestational diabetes, and IUGR. Additionally, one woman had diabetes type I, and three women had diabetes type II before pregnancy. Forty-seven women finished their pregnancies without complications development.”
In order to confirm that the chosen sample size was sufficient, the power of the study was calculated. The calculated power of the study was greater than 0.8, and this sentence was added in the revised Materials and Methods section (Page 12 in the revised Manuscript):
“The calculated power of the study was higher than 0.8.”
Reviewer #2 comment 19: The most used methodology ===without reference(LN/413-481--etc )
Answer: Thank you for the comment.
Serum total cholesterol, triglyceride, high-density lipoprotein (HDL) cholesterol, and apolipoprotein A-I (apoA-I) levels were measured by commercial tests (Beckman Coulter, Brea, California, USA), on an automated biochemical analyzer Beckman AU 480 (Beckman Coulter, Brea, California, USA), while LDL cholesterol levels were calculated according to the Friedewald equation (Reference: Friedewald, W.T.; Levy, R.I.; Fredrickson, D.S. Estimation of the concentration of low‑density lipoprotein cholesterol in plasma, without use of the preparative ultracentrifuge. Clin Chem 1972, 18(6), 499-502. doi: 10.1093/clinchem/18.6.499).
Serum cholesterol precursors (desmosterol, 7-dehydrocholesterol, and lathosterol) and cholesterol absorption markers (campesterol, β-sitosterol), i.e., NCSHDL were quantified by liquid chromatography-tandem mass spectrometry (LC-MS/MS), as previously reported (Reference: Vladimirov, S.; Gojković, T.; Zeljković, A.; Jelić-Ivanović, Z.; Spasojević-Kalimanovska, V. Determination of non-cholesterol sterols in serum and HDL fraction by LC/MS-MS: Significance of matrix-related interferences. J Med Biochem 2020, 39(3), 299-308. doi: 10.2478/jomb-2019-0044).
LCAT and CETP activities were determined as described by Asztalos et al. (Reference: Asztalos, B.F.; Swarbrick, M.M.; Schaefer, E.J.; Dallal, G.E.; Horvath, K.V.; Ai, M.; Stanhope, K.L.; Austrheim-Smith, I.; Wolfe, B.M.; Ali, M.; Havel, P.J. Effects of weight loss, induced by gastric bypass surgery, on HDL remodeling in obese women. J Lipid Res 2010, 51(8), 2405-2412. doi: 10.1194/jlr.P900015).
Serum paraoxonase 1 (PON1) activity was measured as previously described by Richter and Furlong (Reference: Richter, R.J.; Furlong, C.E. Determination of paraoxonase (PON1) status requires more than genotyping. Pharmacogenetics 1999, 9(6), 745-753).
Reviewer #2 comment 20: There is no reference for the statistical analysis. Write as Table(1):----------------and Fig.(1):----------etc---apply for all
Answer: We appreciate your suggestion.
Each table in the manuscript is followed by a corresponding footnote where all statistical tests applied to the data presented in the given table are listed, as follows:
Table (1): Paired-Samples T-Test; Wilcoxon Test; Independent-Samples T Test; Mann-Whitney U Test
Table (2): Wilcoxon Test; Mann-Whitney U Test
Table (3): Paired-Samples T-Test; Wilcoxon Test; Independent-Samples T Test; Mann-Whitney U Test
Reviewer #2 comment 21: The authors did not mention any thing about all of the followings:
a- The recorded clinical signs observed during running this work
Answer: Thank you for pointing this out. The diagnosis of preeclampsia was set following the American College of Obstetricians and Gynecologists (ACOG) criteria (Reference: The American College of Obstetricians and Gynecologists. ACOG Practice Bulletin No. 202: Gestational Hypertension and Preeclampsia. Obstet Gynecol 2019, 133(1), e1 25. doi: 10.1097/AOG.0000000000003018). All women with preeclampsia were diagnosed with new-onset hypertension observed after the 20th week of gestation and with subsequent proteinuria (Page 10 in the revised Materials and Methods section):
“The preeclampsia was diagnosed based on new-onset hypertension observed after the 20th week of gestation followed by subsequent proteinuria.”
b -The recorded mortalities percentages ---if there ---in tables
Answer: No mortality cases were documented during our study.
c-Score lesions --if there
Answer: Since no mortality cases were documented during the study, pathological analysis of the tissue was not performed. Therefore, the previous does not apply to our study.
Reviewer #2 comment 22: LN/504-509---delete as it is previously mention at materials and methods
Answer: We agree that lines 504-509 were previously mentioned in the Materials and Methods section. However, according to the propositions stated in the Instruction for Authors for the International Journal of Molecular Sciences, the manuscript should be followed by Back Matter, which includes a separate Institutional Review Board Statement.
Once again, we give an example of an article that was recently published in the journal that could be used as a respectable example:
Falquet, M.; Prezioso, C.; Ludvigsen, M.; Bruun, J.A.; Passerini, S.; Sveinbjørnsson, B.; Pietropaolo, V.; Moens, U. Regulation of Transcriptional Activity of Merkel Cell Polyomavirus Large T-Antigen by PKA-Mediated Phosphorylation. Int J Mol Sci 2023, 24(1), 895. doi: 10.3390/ijms24010895.
Reviewer #2 comment 23: The most used cited references contained more than 6 authors ---why??? should be 6 at the maximum plus etal with the last ones ---apply for all(ref---8,9,14,19,28,29,30----etc)
Answer: We definitely agree with the Reviewer that when citing, the names of the first six authors are most often mentioned followed by et al.
However, we were just following the proposition of the International Journal of Molecular Sciences. In the Instructions for Authors, it is stated:
“Your references may be in any style, provided that you use the consistent formatting throughout. It is essential to include author(s) name(s), journal or book title, article or chapter title (where required), year of publication, volume and issue (where appropriate) and pagination. DOI numbers (Digital Object Identifier) are not mandatory but highly encouraged.”
Reviewer #2 comment 24: Some cited references need to be more update
Answer: Yes, thank you very much for pointing this out. We have updated some of the references which relate to the understanding of the pathophysiological mechanisms involved in the development of early- and late-onset preeclampsia. We believe that this has significantly improved the scientific value of the manuscript.
Reviewer #2 comment 25: As volume, issue, number and pages ---all are available ----so no need for the link(s)---apply for all
Answer: We appreciate the suggestion, but once again we were following the propositions given by the International Journal of Molecular Sciences:
“DOI numbers (Digital Object Identifier) are not mandatory but highly encouraged.”
Reviewer #2 comment 26: There is no plan for the study area
Answer: Thank you very much for mentioning this.
All pregnant women were recruited at their first antenatal hospital appointment at Obstetrics and Gynecology Clinic “Narodni Front” in Belgrade, as mentioned in the Materials and Methods section (Page 10 in the revised manuscript):
“Women were included in the study at their first antenatal check-up at the Obstetrics and Gynecology Clinic “Narodni Front” (Belgrade, Serbia).”
This clinic has a Department for high-risk pregnancies dedicated to monitoring and managing pregnancies with high-risk for preeclampsia development. Women from all over Serbia are referred to Obstetrics and Gynecology Clinic “Narodni Front” for further monitoring if their pregnancy is found to be high-risk.
Reviewer #2 comment 27: There is no charts or graphics
Answer: Thank you very much for the comment. That's right, no charts and graphics are included in the manuscript. The manuscript is conceptualized so that all data are presented in the form of tables.
Reviewer #2 comment 28: LN/526----why all capitals???
Answer: Thank you. DOI numbers in some journals are listed as such. When citing references, we followed the designation characteristics for each journal. Additionally, this reference was removed in the revised manuscript.
Reviewer #2 comment 29: Some journal names were written abbreviated , while others were not ---why ??? same style should be ---apply for all
Answer: Thank you very much for pointing this out. When citing references, we used NLM Title Abbreviation. The corrections were made where applicable (Pages 13-15 in the revised manuscript).
Reviewer #2 comment 30: LN/134---which is better retardation or restriction
Answer: Thank you very much for the comment.
According to our knowledge, the term restriction is commonly encountered in the scientific literature. Intrauterine growth restriction (IUGR) was defined as the rate of fetal growth below normal regarding the growth potential of a specific infant as per the race and gender of the fetus. An IUGR also applies to neonates born with clinical features of malnutrition and in-utero growth retardation, irrespective of their birth weight percentile (Reference: Sharma, D.; Shastri, S.; Sharma, P. Intrauterine growth restriction: antenatal and postnatal aspects. Clin Med Insights Pediatr 2016, 10, 67-83. doi: 10.4137/CMPed.S40070).
Reviewer #2 comment 31: Early and late preeclampsia (34-37 weeks)---why and what are the prominent features or changes that occurred either to the mother or fetus during this time ????
Answer: Thank you very much for mentioning this.
The pathophysiological mechanisms and crucial differences between the two types of preeclampsia were explained in a detailed manner in the revised Introduction section (Page 2 in the revised manuscript):
„Both early- and late-onset preeclampsia are thought to develop in two stages. The first stage includes poor perfusion of the placenta and consequent placental dysfunction. In response to the inadequate blood supply to the placenta and consequent hypoxia, oxidative stress develops and increased production of free radicals and inflammatory cytokines by the placenta emerges. These molecular mediators of oxidative stress and inflammation lead to the development of generalized maternal endothelial dysfunction in the second stage of the disease development [4,5]. However, the key difference between early- and late-onset preeclampsia lies in the cause and time of placental malperfusion and dysfunction [4]. In the early-onset syndrome, the first stage is the result of poor placentation due to the shallow remodeling of spiral arteries during the first half of pregnancy. Incomplete remodeling of spiral arteries underlies abnormal uteroplacental perfusion, leading to placental oxidative stress and causing the hypersecretion of inflammatory and antiangiogenic factors into the maternal circulation [4]. In women with late-onset preeclampsia, the maturating placenta outgrows the uterus capacity and the supporting maternal functions. In this case, the placenta also becomes under-perfused, with restricted intervillous perfusion, causing placental stress at a later gestational age [4,5]. Placental overgrowth seems to be particularly associated with maternal obesity and large placentas [4]. Hence, both pathways cause placental hypoperfusion and dysfunction in the first stage, leading to the clinically recognized maternal syndrome of preeclampsia in the second stage of disease development. Early-onset syndrome is associated with adverse maternal and neonatal outcomes, such as reduction in placental volume, intrauterine growth restriction (IUGR), abnormal uterine and umbilical artery Doppler evaluation, low birth weight, multiorgan dysfunction, and perinatal death [5,6]. On the other hand, the late-onset disorder is usually associated with a larger placental volume, normal uterine and umbilical artery Doppler evaluation, a normally grown baby with no signs of growth restriction, and significantly more favorable outcomes for the mother and fetus [5,6]. However, late-onset preeclampsia has a higher incidence (2.7% vs. 0.38%) [5,6].“
In our study, pregnant women who developed late-onset preeclampsia were monitored. Late-onset preeclampsia is thought to have an ’intrinsic’ cause of placental dysfunction. This disorder could largely be predisposed by maternal risk factors and seems to share several risk factors with cardiovascular disease, as described in the revised Introduction section (Pages 2 and 3 in the revised manuscript):
„Cardiovascular disease and preeclampsia seem to share several risk factors, such as endothelial dysfunction, hypertension, diabetes, and low-grade systemic inflammatory response [3]. Maternal risk factors, such as chronic vascular diseases, obesity, autoimmune diseases, insulin resistance, and chronic kidney or liver diseases, seem to affect both stages of the disease development. These factors not only slow down the process of placentation (stage 1) but also increase the maternal vascular sensitivity to factors released by the placenta (stage 2). The increased sensitivity of the maternal endothelium to the factors released by the placenta is based on a chronic low-grade inflammation that is present in these women even before pregnancy [4]. Disturbances in lipid and lipoprotein metabolism are already acknowledged as key contributors to atherosclerosis and further cardiovascular disease development. Hence, modulators of lipid metabolism emerge as compelling factors that could be important in preeclampsia development [7].“
Reviewer #2 comment 32: LN/163-172----be more summarize and apply for all
Answer: Thank you for the comment.
Footnotes below the tables were summarized in the revised Results section (Pages 5, 6, and 7 in the revised manuscript).
Reviewer #2 comment 33: LN/228-233----where is the discussion???
Answer: Thank you very much for pointing this out. Following the received criticism, lines 228-233 were deleted in the revised Discussion section (Page 7 in the revised manuscript).
Reviewer #2 comment 34: LN/234-242---this is like introduction not a discussion
Answer: We appreciate the criticism. Lines 234-242 were deleted in the revised Discussion section (Page 7 in the revised manuscript).
Reviewer #2 comment 35: Discussion should be rewritten again and be based upon debating the obtained results with those of the previous investigators results
Answer: Thank you for pointing that out. We certainly agree with the Reviewer's comment. Here we would like to highlight once again that most of the parameters that were examined in our study, according to our knowledge, have not been investigated in women with preeclampsia so far. According to our findings, this study is the first one examining the concentrations of non-cholesterol sterols in the HDL fraction, LCAT, and CETP activities in women with preeclampsia. We believe that it adds significantly to the strength and the scientific value of our manuscript. Some parts of the discussion have been further elaborated, and the results obtained in our study were debated concerning the results obtained in previous studies, wherever possible (Pages 7-10 in the revised manuscript).
Round 2
Reviewer 1 Report
the manuscript can still be improved
this is a retrospective cohort study with sampling at 2 points in longitudinal follow up
there patients with diabetes and there is no analysis for this effect either by stratification sensitivity analysis or adjustments in the statistical methods
statistical methods are not complete not adequately sophisticated for the outcome
can be improved
Author Response
A point-by-point reply follows:
Reviewers’ comments:
Reviewer #1 comment 1: The manuscript can still be improved.
Answer: We appreciate your opinion. Accordingly, we have taken into account all your comments, and we believe that the changes we have made in the re-revised version have significantly improved the scientific value of our manuscript.
Reviewer #1 comment 2: This is a retrospective cohort study with sampling at 2 points in longitudinal follow up.
Answer: We agree with the reviewer’s assessment. Therefore, we modified the line related to the study design in the Materials and Methods section, as follows (Page 10 in the re-revised manuscript):
„In this retrospective cohort study with sampling at two points in longitudinal follow-up, we followed 90 pregnant women throughout gestation longitudinally.“
Reviewer #1 comment 3: There patients with diabetes and there is no analysis for this effect either by stratification sensitivity analysis or adjustments in the statistical methods.
Answer: Thank you very much for the comment.
In the re-revised manuscript, we took into account the diabetic status of the patients. The analysis of covariance (ANCOVA) or non-parametric ANCOVA (Quade method) was used to compare the means or medians between the two study groups: RG and PG, thereby regarding diabetic status as a covariate, as explained in Cangür Åž, Sungur MA, Ankarali H. The methods used in nonparametric covariance analysis. Duzce Medical Journal 2018, 20(1), 1-6. ANCOVA analyzes for differences in response variable among groups, taking into account the variability in the response variable explained by one or more covariates. In the re-revised manuscript, we applied ANCOVA instead of the Independent Samples T-Test for data that were normally or log-normally distributed. If the either normality assumption of the conditional distributions of the response variable or any other assumption for ANCOVA was not satisfied, then Ranked ANCOVA was applied. In the re-revised manuscript ranked ANCOVA, i.e., non-parametric ANCOVA (Quade method) was used to compare the medians between PG and RG for data that did not follow normal distribution, instead of Mann-Whitney U Test. The only covariate we used was the diabetic status of our patients. All subjects with diagnoses of diabetes mellitus type I, diabetes mellitus type II, and gestational diabetes were considered positive for diabetes.
Corresponding corrections were made in the re-revised Statistical analysis section (Page 12 in the re-revised Materials and Methods section):
“The analysis of covariance (ANCOVA) or non-parametric ANCOVA (Quade method) was used to compare the means or medians between the two study groups: RG and PG, thereby regarding diabetic status as a covariate.”
All changes in statistical significance (differences in triglycerides levels in the 3rd trimester and campesterol levels in the 2nd trimester), which were observed after the application of the ANCOVA and Quade method, were entered in the re-revised manuscript (Pages 5 and 6 (Table 1 and Table 2) in the re-revised Results section). Corresponding changes in the interpretation of results are also included in the re-revised manuscript (Page 4 in the re-revised Results section):
“Even though an increase in total cholesterol concentration between 2nd and 3rd trimesters is observed in both groups, we did not find significant changes in LDL and HDL cholesterol concentrations between trimesters in PG. In both study groups, a significant increase in triglycerides concentration between the 2nd and 3rd trimesters was observed (p<0.001). The concentration of HDL cholesterol was significantly lower in the 2nd trimester in PG compared to RG (p<0.05). Women in PG had significantly higher triglycerides levels in the 2nd trimester (p<0.05), while triglycerides concentrations in the 3rd trimester were also higher in the PG but of borderline significance (p=0.051) (Table 1).”
Reviewer #1 comment 4: Statistical methods are not complete not adequately sophisticated for the outcome.
Answer: We appreciate the Reviewer’s feedback.
As suggested by the Reviewer, we have made significant changes in the applied statistical methods. In the re-revised manuscript, the analysis of covariance (ANCOVA) or non-parametric ANCOVA (Quade method) was used to compare the means or medians between the two study groups: RG and PG, thereby regarding diabetic status as a covariate, instead of Independent Samples T-Test and Mann-Whitney U Test.
Reviewer #1 comment 5: Comments on the Quality of English Language: can be improved.
Answer: Thank you very much for pointing this out. The manuscript was proofread by an official English-speaking person. Corresponding changes were entered throughout the manuscript.
Reviewer 2 Report
AS the authors have responded to my corrections , so the decision is accepted for publishing
Author Response
Reviewer #2 comment 1: As the authors have responded to my corrections, so the decision is accepted for publishing.
Answer: Thank you for your time and all your comments; we believe that your suggestions, which we adopted, significantly improved the scientific value of our manuscript.
Round 3
Reviewer 1 Report
the manuscript is much better the authors have been responsive
During the response the authors state that
all participants were actually sampled sooner in pregnancy and this is not reported in addition
However, for the purpose of this manuscript, we concentrated only on the two points, to simplify the presentation of the observed data and results. Two points were targeted – the second trimester of pregnancy, before the appearance of clinical signs of preeclampsia, when important changes in the metabolism of HDL are expected (increase in HDL-C concentration in pregnancies without complications), and the late third trimester, when the diagnosis of late-onset preeclampsia was already set in the PG.
why is this? simplification doesn't seem justified
I should think this information is precious for their cause and for the interest of others most persons arrive at their Obstetrician early in pregnancy and this may be mechanistically meaningful the literature is wanting because of a lack of sampling throughout pregnancy
Author Response
A point-by-point reply follows:
Reviewers’ comments:
Reviewer #1 comment 1: The manuscript is much better the authors have been responsive.
Answer: Thank you for your time and all your comments; we believe that your suggestions, which we adopted, significantly improved the scientific value of our manuscript.
Reviewer #1 comment 2: During the response the authors state that all participants were actually sampled sooner in pregnancy and this is not reported in addition:
However, for the purpose of this manuscript, we concentrated only on the two points, to simplify the presentation of the observed data and results. Two points were targeted – the second trimester of pregnancy, before the appearance of clinical signs of preeclampsia, when important changes in the metabolism of HDL are expected (increase in HDL-C concentration in pregnancies without complications), and the late third trimester, when the diagnosis of late-onset preeclampsia was already set in the PG.
Why is this? Simplification doesn't seem justified.
I should think this information is precious for their cause and for the interest of others. Most persons arrive at their Obstetrician early in pregnancy and this may be mechanistically meaningful. The literature is wanting because of a lack of sampling throughout pregnancy.
Answer: We agree with the reviewer’s assessment that the literature lacks data on longitudinal monitoring of changes during pregnancy. Therefore, pregnant women included in our study were monitored from the beginning to the end of pregnancy, i.e., from the first trimester until the delivery. The initial assessment of the basic clinical and laboratory parameters in all subjects was also done early in the course of pregnancy, i.e., in the first trimester. The results of that analysis were statistically processed, and the comparison with the results obtained in the second trimester is shown in the following table:
Table S1. Changes in lipid profile parameters in RG and PG in the 1st and 2nd trimester
|
1st trimester |
2nd trimester |
p1 |
p2 |
|||
|
|
RG (N=70) |
PG (N=20) |
RG (N=70) |
PG (N=20) |
||
|
WG |
12.8 ± 0.80 |
12.7 ± 0.16 |
23.3 ± 0.83 |
23.4 ± 0.99 |
|
|
|
Age, years |
32.0 ± 5.56 |
33.1 ± 4.26 |
|
|
|
|
|
MAP, mmHg |
85.9 ± 11.59 |
92.4 ± 9.01dg |
82.5 ± 11.51 |
89.8 ± 9.64dg |
<0.05 |
0.164 |
|
BMIb, kg/m2 |
23.6 (21.1-27.5) |
27.5 (24.2-31.1) |
25.4 (23.6–30.0) |
28.4eg (26.4–32.7) |
<0.001 |
<0.001 |
|
Weight gainb, kg |
2.15 (1.27-4.07) |
3.00 (0.25-4.40) |
5.40 (3.72–7.02) |
4.70 (2.82–6.75) |
<0.001 |
<0.05 |
|
Weight gainb, % |
4.00 (2.00-6.00) |
4.00 (2.25-5.90) |
7.00 (5.00–9.75) |
5.50 (4.00–8.75) |
<0.001 |
<0.05 |
|
TC, mmol/L |
5.33 ± 1.070 |
5.46 ± 0.762dg |
6.81 ± 1.365 |
6.61 ± 1.236 |
<0.001 |
<0.001 |
|
HDL-C, mmol/L |
1.77 ± 0.336 |
1.90 ± 0.587 |
2.11 ± 0.386 |
1.88 ± 0.363dg |
<0.001 |
0.885 |
|
LDL-C, mmol/L |
2.94 ± 0.849 |
2.81 ± 0.633 |
3.81 ± 1.148 |
3.61 ± 1.104 |
<0.001 |
<0.001 |
|
TGa, mmol/L |
1.27 (1.18-1.38) |
1.53 (1.27-1.85) |
1.86 (1.73–2.00) |
2.33dg (2.00–2.71) |
<0.001 |
<0.001 |
RG – risk group; PG – preeclampsia group; WG – week of gestation; BMI – body mass index; MAP – mean arterial pressure; TC – total cholesterol; HDL-C – high density lipoprotein cholesterol; LDL-C – low density lipoprotein cholesterol; TG – triglycerides
Data are shown as mean ± standard deviation; a geometric mean (95th CI); b median (interquartile range)
p1 – Paired-Samples T-Test for risk group; p2 –for preeclampsia group
p1c – Wilcoxon Test for risk group; p2c –for preeclampsia group
Difference significantly different from the risk group: d Analysis of covariance (ANCOVA); e Non-parametric ANCOVA (Quade method)
f p<0.001; g p<0.05
Following the reviewer's advice, the results of this analysis will be added in the form of supplementary material (Table S1. Changes in lipid profile parameters in RG and PG in the 1st and 2nd trimester) during the submission of the re-revised manuscript. We believe that these results might be interesting and significant for the scientific public. The appropriate change was made in the re-revised manuscript (Page 13):
„Supplementary Materials: Table S1: Changes in lipid profile parameters in RG and PG in the 1st and 2nd trimester“
The usual biochemical assessment in pregnancy includes complete blood count analysis, analysis of changes in glucose metabolism (glycemia assessment), total protein and albumin concentration, creatinine concentration, chemical analysis of urine, and microscopic examination of urine sediment. Even though pregnancy is a condition that is accompanied by intense changes in lipid metabolism, up to now, there were no guidelines that would indicate the need for regular monitoring of lipid status in pregnant women. This manuscript presents some of the observations that resulted from a project entitled High-density lipoprotein metabolome research to improve pregnancy outcome – HI-MOM, grant number #7741659, funded by the Science Fund of the Republic of Serbia. One of the indirect aim of our Project is to point out the importance of determining and monitoring the lipid status in pregnant women primarily through the analysis of the basic parameters of the lipid profile and further through some specific components of the metabolome of HDL particles.
The established fact of lipids profile parameters change during uncomplicated pregnancy is increase in HDL-C concentrations in the 2nd trimester. At the other side, a number of studies noticed that there were no significant changes in HDL cholesterol concentration between the first and second trimester in women who developed preeclampsia. It is now clear that the role of HDL particles goes far beyond the reverse cholesterol transport and cholesterol concentration. This lipoprotein particle has very complex composition with more than 100 different proteins, cholesterol, non-cholesterol sterols (NCSs), glycerophospholipids, triglycerides and other lipid compounds and indicates many physiological functions. As apparently, the most significant changes in the metabolism of HDL particles occur in the second trimester of pregnancy, we decided to further analyze the composition and functionality of these particles, i.e., HDL lipidome, in this pregnancy point instead during the whole follow-up. The late third trimester was taken as a comparison point when the diagnosis of late-onset preeclampsia was already set in the PG.
Round 4
Reviewer 1 Report
authors are responsive and the information is additive
in the section 'study design' needs to be revised to let the readers know you are reporting all trimesters now for at least some of your analysis
Author Response
Reviewers’ comments:
Reviewer #1 comment 1: Authors are responsive and the information is additive.
Answer: We are very grateful for the reviews provided by the external reviewer of this manuscript. The comments are encouraging and the reviewer appears to share our judgement that this study and its results are clinically important.
Reviewer #1 comment 2: In the section 'study design' needs to be revised to let the readers know you are reporting all trimesters now for at least some of your analysis.
Answer: Thank you very much for pointing this out. Following the reviewer’s suggestion we have made changes at several sections in the manuscript.
As advised by the reviewer we have provided more additional information about the longitudinal follow-up of basic lipid profile parameters in the Study design section (Page 10 in the re-revised manuscript):
“In this retrospective cohort study with sampling at three (for basic lipid profile parameters) or two points (for NCSs and HDL functionality markers) in longitudinal follow-up, we monitored 90 pregnant women throughout gestation longitudinally.”
Accordingly, we have made changes in the Sample collection section also, explaining in detail when blood sampling was performed for each parameter tested (Page 11 in the re-revised manuscript):
“Blood was sampled at two points – in the 2nd trimester (22–25 weeks of gestation), before the appearance of clinical signs of preeclampsia, and in the 3rd trimester (35–38 weeks of gestation), when clinical signs of late-onset preeclampsia were manifested, for the analysis of NCSs and HDL functionality markers. In addition, for the analysis of the basic parameters of the lipid profile, blood was also sampled at the beginning of pregnancy, i.e. in the first trimester (12-14 weeks of gestation).”
A corresponding correction has been in the Results section to let readers know where they could look for the information on the additional comparison of basic lipid profile parameters between the 1st and 2nd trimester (Page 4 in the re-revised manuscript):
“Additional comparisons of basic clinical and lipid profile parameters in the 1st and 2nd trimesters can be found in the supplementary material (Table S1).”